# SA-β-Gal in Kidney Tubules as a Predictor of Renal Outcome in Patients with Chronic Kidney Disease

**DOI:** 10.3390/jcm13020322

**Published:** 2024-01-06

**Authors:** Pasquale Esposito, Daniela Picciotto, Daniela Verzola, Giacomo Garibotto, Emanuele Luigi Parodi, Antonella Sofia, Francesca Costigliolo, Gabriele Gaggero, Valentina Zanetti, Michela Saio, Francesca Viazzi

**Affiliations:** 1Department of Internal Medicine, University of Genova, 16132 Genova, Italy; pasquale.esposito@unige.it (P.E.); daverz@libero.it (D.V.); valentinazanetti94@gmail.com (V.Z.); francesca.viazzi@unige.it (F.V.); 2Division of Nephrology, Dialysis and Transplantation, IRCCS Ospedale Policlinico San Martino, 16142 Genova, Italy; daniela.picciotto@hsanmartino.it (D.P.); antonella.sofia@hsanmartino.it (A.S.); francesca.costigliolo@hsanmartino.it (F.C.); michela.saio@virgilio.it (M.S.); 3UO Anatomia Patologica, IRCCS Ospedale Policlinico San Martino, 16142 Genova, Italy; gabriele.gaggero@hsanmartino.it

**Keywords:** senescence, cell, chronic kidney disease, kidney tubules

## Abstract

Cellular senescence has emerged as an important driver of aging and age-related disease in the kidney. The activity of β-galactosidase at pH 6 (SA-β-Gal) is a classic maker of senescence in cellular biology; however, the predictive role of kidney tissue SA-β-Gal on eGFR loss in chronic kidney disease (CKD) is still not understood. We retrospectively studied the expression of SA-β-Gal in kidney biopsies obtained in a cohort [*n* = 22] of incident patients who were followed up for 3 years as standard of care. SA-β-Gal staining was approximately fourfold higher in the tubular compartment of patients with CKD vs. controls [26.0 ± 9 vs. 7.4 ± 6% positive tubuli in patients vs. controls; *p* < 0.025]. Tubular expressions of SA-β-Gal, but not proteinuria, at the time of biopsy correlated with eGFR loss at the follow up; moreover, SA-β-Gal expression in more than 30% of kidney tubules was associated with fast progressive kidney disease. In conclusion, our study shows that SA-β-Gal is upregulated in the kidney tubular compartment of adult patients affected by CKD and suggests that tubular SA-β-Gal is associated with accelerated loss of renal function.

## 1. Introduction

Cellular senescence is now recognized as a major driver of aging and age-related disease in the kidney [1,2,3]. Cellular senescence has been defined as an irreversible growth arrest associated with functional and morphological changes, including gene transcription, chromatin organization, and distinct protein secretion [1,2]. In vitro senescent cells are characterized by increased cell size and enzymatic activity of the lysosomal hydrolase senescence-associated β-galactosidase [SA-β-Gal] [4], and upregulation of the cyclin-dependent kinase inhibitors p16^INKA^ and/or p21^cip1^ and prosurvival pathways to resist apoptosis [1,2]. Moreover, senescent cells may develop a distinctive secretory phenotype consisting of various proinflammatory molecules, metalloproteases, and growth factors, which is known as “senescence-associated secretory phenotype” [SASP] [4,5]. Cell senescence is produced both as a physiological response during development, or after multiple stressors, such as genotoxic injury, oncogene activation, hypoxia, cellular stress, mitochondrial dysfunction, or nutrient deprivation [3] The consequences of accelerated cell senescence in kidney disease are twofold. Because of cell cycle arrest, senescence is thought to cause a loss of the capacity of tissue repair; this is especially relevant to cells with low replication rates. In addition, senescent cells produce proinflammatory and matrix-degrading molecules in the SASP, which are thought to accelerate glomerulosclerosis and interstitial fibrosis [5,6]. As a matter of fact, different experimental models of CKD identified that accumulation of senescent cells arrested later the G2/M phase triggers the SASP phenotype [7]. This is more pronounced in tubular epithelial cells and affects the tubulointerstitial space [8]. The definition of cell senescence in the kidney has practical implications in consideration of the new “senolytic” therapies which, by “weeding out” accumulated senescent cells, could potentially recover tissue function [3,9]. However, it is currently complicated to define unique markers which permit both the identification and quantification of senescent cells [9]. Available studies are based on the presence/absence of cell-cycle arrest markers [3,9]; however, this may not be sufficient, as the increase in nuclei positively immunostaining for CDK4/6 inhibitors such as p16^INKA^ does not always coincide with the upregulation of p16^INKA^ expression in vivo [6,10,11]. Even if there are many potential markers of senescence, their specificity is likely influenced by a number of factors, including cell type and developmental stage [11]; for this reason, it has been proposed that the determination of tissue senescence needs the inclusion of three approaches: staining the cell-cycle arrest markers (SA-β-Gal or lipofuscin), quantifying the effects of senescence-associated protein release, and excluding coexisting cellular proliferation [11]. Recently, in addition to the study of CDK4/6 inhibitors, other phenotypic markers, among which are those of altered mitochondrial dynamics and the secretome, and telomere shortening, have been proposed to identify senescent cells [11,12].

SA-β-Gal is a lysosomal hydrolase, which cleaves terminal β-d-galactose residues [13]. The activity of SA-β-Gal is a classic maker of senescence in cellular biology [4]. The regulation of SA-β-Gal activity is independent of DNA synthesis and reflects changes in cell function that accompany senescence [4]. SA-β-Gal detection is easy and relatively fast to perform on frozen samples of kidney [9,13,14] and has been proposed as a screening test for subsequent use of more complex and complementary techniques [9]. However, SA-β-Gal is not a completely specific marker of cellular senescence [15]. In patients with diabetic nephropathy, we observed that SA-β-Gal staining in the tubular compartment was directly related to the nuclear + cytoplasmic expression of p16^INK4A^, but not to nuclear p16^INK4A^. When we examined the SA-β-Gal and p16^INK4A^ co-expression, some tubule cells showed a complete co-expression of these two markers, while others showed an isolated expression of SA-β-Gal or p16^INK4A^ [13]. This suggests that the aging phenotypes are differently expressed at the same time in the same tissue. In addition, the predictive role of SA-β-Gal on eGFR loss is still not known, since in studies conducted with small cohorts of patients with glomerular diseases, telomere shortening activity [16] and p16^INKA^ expression, but not SA-β-Gal, were related to subsequent eGFR loss [17]. In this study, we tested the hypothesis that elevated SA-β-Gal activity in kidney biopsies is a predictor of subsequent eGFR loss in patients with CKD. With this in mind, we retrospectively studied the expression of SA-β-Gal in kidney biopsies obtained from a cohort of incident kidney patients who were followed up for 3 years as standard of care. 

## 2. Materials and Methods

### 2.1. Tissue Samples

We reviewed computerized results of renal biopsies obtained at the Department of Internal Medicine, Nephrology Division, Genoa University, and IRCCS AOU San Martino, Genoa, Italy, from May 2007 to June 2011, in which SA-β-Gal staining was available. The review included subjects aged 18 years or older, with or without type 2 diabetes, with a urinary protein excretion of 150–10,000 mg/day, and an eGFR of 25–120 mL/min per 1.73 m^2^. All biopsies were performed not for research but for clinical reasons. The results obtained for FSGS have been recently published [17]. This study was part of a larger study on cell senescence and the kidney sponsored by the Ministero dell’Università e della Ricerca Scientifica [MIUR] [Progetti Finalizzati per la Ricerca di Base-FIRB 2002] and approved by the Ethical Committee of the Azienda Ospedale San Martino (Ref. N.72006) [14,18]. The procedures were in accordance with the Declaration of Helsinki. All kidney biopsy specimens were analyzed by the same renal pathologist. Estimated GFR was assessed by CKD-EPI formula [18]. One hundred and fifteen consecutive kidney biopsy cases in which SA-β-Gal activity was available were assessed for eligibility [Figure 1]. 

Seventy-three cases with eGFR < 25 mL/min or insufficient tissue availability were excluded from further analysis. After an initial evaluation, 20 cases resulted in being lost at the follow up [36 months]. At the end of the follow up, 22 biopsies were available, including IgA nephropathy [*n* = 5], FSGS [*n* = 5], membranous glomerulonephritis [*n* = 5], minimal change disease [*n* = 1], chronic glomerulonephritis [*n* = 1], hypertensive nephroangiosclerosis [*n* = 2], stage II lupus nephritis [*n* = 1], and diabetic kidney disease [*n* = 2]. Baseline clinical and demographic characteristics of CKD patients are shown in Table 1.

The average urine protein excretion was 2.7 ± 1.0 g/day [median 1.9, IQR 0.75, 4.42]; urine protein excretion was 150–200 mg/day for 4 cases, and 9 subjects presented with nephrotic (>3 g/day) proteinuria. The mean eGFR was 80 ± 5.7 mL/min 1.73 m^2^ and it was less than 60 mL/min/1.73 m^2^ for 11 participants (38%). Routine clinical visits, including clinical labs, proteinuria, and eGFR estimations, were carried out basally, after 1–6 months from biopsy, and then at least annually for 3 years. This timeline represents standard of practice. Twenty-two subjects received single or double RAS blockade, and sixteen subjects received statins. Thirteen subjects received steroids/immunosuppressive therapy. Four patients (28% of cases) doubled their serum creatinine and two subjects reached ESRD at the end of the follow up. Two patients deceased during the follow up. Normal portions of kidney tissue from nephrectomies for renal carcinoma [*n* = 13, 8 males and 5 females, 60 ± 4 years], were examined as a control group. In these subjects, a physical examination and screening biochemical tests of renal, hepatic, hematological, and metabolic function (thyroid function and fasting plasma glucose) were unremarkable.

### 2.2. SA-β-Gal Activity on Cryopreserved Kidney Biopsies

Biopsies were embedded in OCT Compound (Tissue-Tek OCT compound by Sakura, Torrance, CA, USA), flash-frozen in cold isopentane (Merck Group, Milan, Italy), and stored at −80 °C. SA-b-Gal staining was assessed 24 h from the freezing of the sample. The frozen tissues were sectioned by a −20 °C Cryostat (5 µm) and fixed in 0.5% glutaraldehyde (Merck Group) for 10 min at room temperature. SA-β-Gal staining was performed as described by Dimri et al. [4,14]. All reagents were purchased by Merck Group, Milan, Italy. Slides were observed under a microscope, and SA-β-Gal staining was detected at 20× magnification as blue precipitate in tubuli. SA-β-Gal quantification was performed by ImageJ 1.X software and expressed as the percentage of SA-β Gal positive tubuli as a function of total numbers of tubuli [13]. In brief, for analysis, the captured images were converted to gray scale, and the threshold parameters were adjusted to ensure that only positive tubuli were evaluated [13]. 

### 2.3. Statistical Analysis

Data are presented as means ± SEM. Parameters not normally distributed were logarithmically transformed for statistical analysis. While proteinuria, cholesterol, triglycerides, and LDL were not normally distributed, SA-β-Gal passed normality tests. Comparisons between groups were made by ANOVA. Comparisons of proportions were made using the χ^2^-test or Fisher’s exact test whenever appropriate. Relationships between parameters were analyzed using simple regression analysis or the Spearman test, as required. Multivariate logistic regression analysis was performed to assess which parameters significantly related to each other in the simple regression analysis were independent predictors of eGFR loss at the follow up. The Statview statistical package [Cary, NC, USA] was used for the analysis. A 2-tailed *p* value < 0.05 was considered statistically significant.

## 3. Results

### 3.1. SA-β-Gal Staining in the Tubular Compartment

The SA-β-Gal signal was predominantly observed in tubular cells, and it was also only weakly noticed in glomerular parietal cells, podocytes, and vascular endothelial cells, in accordance with findings previously observed [13]. Tubular SA-β-Gal staining was ~2.6-fold higher in biopsies from patients with CKD patients vs. controls [26 ± 4 vs. 10 ± 6% positive tubuli in patients vs. controls; *p* < 0.025] [Figure 2A,B]. 

About 63% of biopsies had >10% of tubuli expressing SA-β-Gal. Mean eGFR was 80 ± 5.7 mL/min/1.73 m^2^ at baseline, 75 ± 5 mL/min/1.73 m^2^ at year 1, 70 ± 6.1 mL/min/1.73 m^2^ at year 2, and 69 ± 6.7 mL/min/1.73 m^2^ at year 3 [Figure 3]. Mean change in eGFR from baseline to year 3 was −3.54 [95% CI, −0.598 to −6.48] mL/min/1.73 m^2^/year.

### 3.2. Clinical Correlates of SA-β-Gal Expression in Kidney Tubules

Table 2 shows the associations between SA-β-Gal expression and clinical findings in subjects with CKD. According to univariate linear regression analysis, SA-β-Gal was not related to age, BMI, serum creatinine, eGFR, PAS, PAD, total cholesterol, LDL cholesterol, and Triglycerides. Surprisingly, in this cohort of non-selected patients, SA-β-Gal expression was only as a trend [r = 0.27, *p* = NS] related to logproteinuria at the time of the biopsy.

### 3.3. Predictors of eGFR Loss at 36 Months

In univariate analysis, eGFR loss at 36 months was predicted by SA-β-Gal expression in kidney tubules but not by baseline logproteinuria [Table 3, Figure 4 and Figure 5]. No association was observed between the use of immunosuppressive therapy, or other clinical parameters as well, and eGFR loss. Upon visual inspection of the regression line [Figure 4], a tubular SA-β-Gal percent expression greater than 30% was associated with fast (>3.5 mL/min/year) progression. 

## 4. Discussion

Cell senescence is accelerated in several apparently different kidney diseases, including hypertensive disease, acute kidney injury (AKI), IgA nephropathy, focal segmental glomerulosclerosis, diabetic nephropathy, and chronic glomerulonephritis [2,3], which suggests a common underlying mechanism. SA-β-Gal staining has been proposed for initial cell senescence screening, since this enzyme activity is upregulated by senescence in several tissues. However, scarce information is available on SA-β-Gal in human kidney biopsies.

Two matters are addressed in this study which deserve discussion. The first is the overexpression of SA-β-Gal in the tubular compartment of a large percentage of kidney biopsies at the time of diagnosis [~63% of biopsies had >10% of tubuli expressing SA-β-Gal], a finding that indicates that an increase in SA-β-Gal activity in the tubular compartment is an early event in different kidney diseases. The second is the association between the tubular overexpression of SA-β-Gal and eGFR loss at the follow up, a finding which is in accordance with the hypothesis that SA-β-Gal activity in kidney biopsies at the time of diagnosis is predictive of subsequent loss of function. 

In addition to its recognized role as a marker of cell senescence [4,13], SA-β-Gal, at high enzyme activity, can injury the basement membranes through hydrolysis of proteoglycans glycosidic bonds [19,20]. Even as low as 10% of senescent cells in tissues appears to be sufficient to cause damage and dysfunction in studies with in vitro and in vivo models [21]. However, no information is available in humans on the percentage of tubular SA-β-Gal expression which is associated with kidney damage. In our study, contrarywise to in vitro studies [21], a low (10–20%) expression of SA-β-Gal in kidney tubules was associated with limited or no eGFR loss at the follow up. From the study of the regression line between SA-β-Gal activity and a loss of eGFR, a “fast” progression rate, i.e., an eGFR loss > 3.5 mL/min year, was observed when the percentage of SA-β-Gal in tubuli was >30%. Our data therefore suggest that in human kidney disease, the predictive cut-off for tubular SA-β-Gal detection is greater than suggested by in vitro studies. Clearly, our data are preliminary, and ad hoc analyses need to be performed in larger studies.

In addition, the association between SA-β-Gal activity in kidney tubules and accelerated disease progression, in addition to being statistically significant, was not really strong, and the increase in SA-β-Gal activity explained only about 20% (r^2^ = 0.19) of changes in eGFR 3 years after the biopsy. We have to consider that some of the results reported here may have been the resultant not of “chronic” but of “acute” cell senescence. As a matter of fact, cell senescence may develop during an acute response after injury as a mechanism of tissue repair [21]; this kind of “acute senescence” is a tightly controlled process which participates in repair mechanisms and limits fibrosis. In contrast, in chronic diseases, senescent cells accumulate in the kidney in response to a variety of stressors such as metabolic stress, telomere shortening [22], oncogenic mutations, inflammation, and mitochondrial dysfunction [23]. These stressors promote cell-cycle arrest via pathways either dependent or independent of the DNA damage response. Differently from the “acute senescence” setting, in chronic diseases, senescent cells are scarcely removed by apoptosis or immune clearance and are increasingly considered to be mediators of disease progression. In a previous study of FSGS patients [17], while a tubular p16^INK4A^ expression was associated with tubule atrophy, tubular SA-β-Gal staining was associated with interstitial fibrosis. These findings suggest that SA-β-Gal and p16^INK4A^ play different roles or that they are an expression of different mechanisms leading to chronic tubulointerstitial changes. However, considering the non-impressive predictive association between SA-β-Gal and disease progression, one could argue that its role as a general marker of senescence [10] needs to be reassessed.

A pressure-dependent association between increased cellular senescence and hypertension has been demonstrated in hypertensive rats [24]. Angiotensin II, a major player in hypertension and its complications, is increasingly recognized as a mediator of cell senescence [25]; in addition, antihypertensive treatment blunts hypertensive-related cell senescence in deoxycorticosterone acetate-salt-treated rats [26]. However, it is currently unknown if removal of senescent cells does indeed treat hypertension or protects the body from hypertensive organ damage in humans. In our study, we were not able to observe an association between tubular SA-β-Gal expression and blood pressure levels; however, almost all subjects were on treatment with RAAS inhibitors, which might have modified the expression or activity of SA-β-Gal. In addition, we could not find an association between age and SA-β-Gal activity in kidney tubules, a finding in keeping with the adult population studied. 

The progressive decrease in eGFR results in an accumulation of metabolites interfering with many cellular pathways, including cellular senescence [27]. In our study, as also previously observed in a cohort of patients with diabetic nephropathy [14], eGFR was not related to tubular SA-β-Gal, suggesting that renal failure per se is not primarily responsible for senescence activation in initial CKD stages. As a matter of fact, we observed an upregulation of SA-β-Gal activity also in biopsies from subjects with normal or subnormal GFR, suggesting either that the acceleration of senescence is an early change and/or that conditions different from a decreased eGFR may account for SA-β-Gal upregulation. 

Glucose reabsorption by the proximal tubule has been proposed as one of the mechanisms leading to accelerated cell senescence is in kidney tubules [17,26]. The process of continuous glucose reabsorption by proximal tubules is now considered one of the mechanisms of tubulointerstitial damage and CKD progression both in diabetic and in non-diabetic disease [28,29]. In previous studies, we observed also that human proximal tubule cells exposed to high glucose enter into a senescent, non-replicative state in which they are metabolically active but do not respond to mitogenic stimuli [17]. Cells are characterized by enhanced expression of senescence markers, including SA-β-Gal, and different sets of genes, including negative regulators of the cell cycle [mainly p16^INKA^]. Interestingly, these changes can be prevented by pre-treatment with phloryzin, an inhibitor of SGLT action ([26], unpublished). 

In a previous study [13] of patients with diabetic nephropathy, we observed that SA-β-Gal expression in kidney tubules was inversely related to BMI, suggesting a role of metabolic stress on inducing premature cell senescence. However, we could not observe this association in patients with non-diabetic CKD, which suggests an interaction between obesity and diabetic milieu that accelerates cell senescence. 

Finally, we could not find an association between proteinuria and SA-β-Gal in tubular cells. A possible explanation is that in addition to proteinuria, several other stressors [6,9] may cooperate to increase SA-β-Gal activity in kidney disease.

## 5. Conclusions

In summary, our study shows that SA-β-Gal is upregulated in the kidney tubular compartment of adult patients with CKD of different etiologies, and that SA-β-Gal expression > 30% in kidney tubuli is associated with accelerated loss of renal function. Therefore, our results are in accordance with the hypothesis that SA-β-Gal overexpression is associated with accelerated loss of eGFR in CKD. This study provides a clinical basis for the use of SA-β-Gal as a marker of kidney senescence in CKD, a common clinical condition that is still associated with an unfavorable prognosis. However, our results cast some doubt on the use of SA-β-Gal as a unique screening marker of cell senescence in the human kidney. Our study has several strengths, since it includes a real-life renal biopsy sample cohort from incident patients with quite a long follow up. Even if the cohort studied may appear to be heterogeneous, it is important to underline that cell senescence is accelerated in apparently different kidney diseases, including hypertensive disease [24], AKI [7], IgA nephropathy [15], focal segmental glomerulosclerosis [16], diabetic nephropathy [13], and chronic glomerulonephritis [30]. All of these different diseases have in common the upregulation of cell senescence markers (mainly p16^INKA^ and p27) in tubules, which correlates with IFTA, suggesting that kidney tubule senescence has common underlying mechanisms. However, our study has several limitations. This study is retrospective and the cohort studied is small and possibly statistically underpowered to detect all of the existing clinical associations of tubular cell senescence. In addition, all patients received RAAS and/or β-blockers and many were also on treatment with statins, drugs which are known to possess a “senostatic” action, i.e., they can “downregulate” the senescent phenotype [31,32,33,34,35]. The stimulation of angiotensin type 1 (AT1) receptors, as well as of mineralocorticoid receptors, causes oxidative stress and vascular inflammation, which may lead to arterial stiffness and vascular aging [34,35]. Drugs of common practice in nephrology, such as RAAS and mineralocorticoid blockades, atorvastatin, rosiglitazone, and omega-3 fatty acids, are being re-evaluated for their anti-senescence action. For all of these reasons, the findings obtained here need to be better considered as hypothesis-generating rather than conclusive, and need to be confirmed in multicentric, prospective, controlled studies.

## Figures and Tables

**Figure 1 jcm-13-00322-f001:**
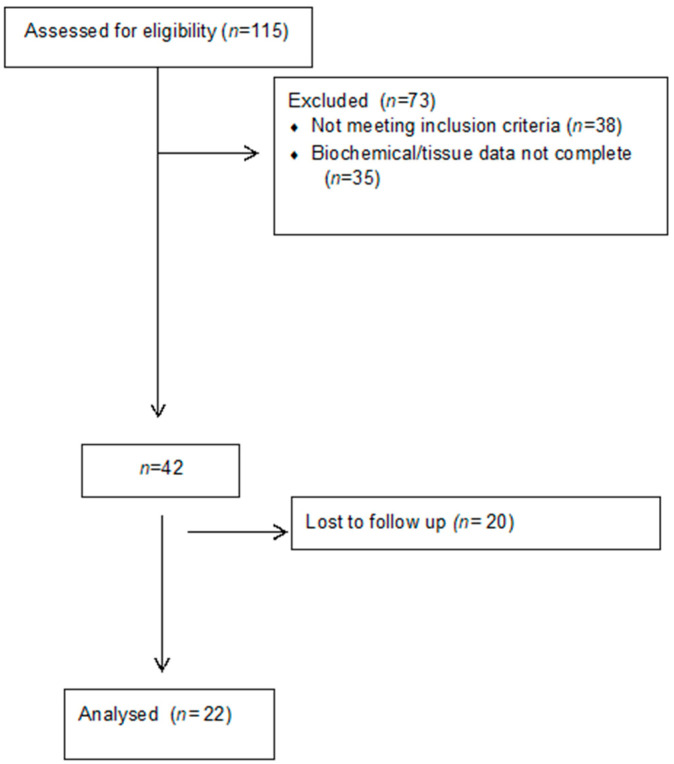
Diagram of the study flow.

**Figure 2 jcm-13-00322-f002:**
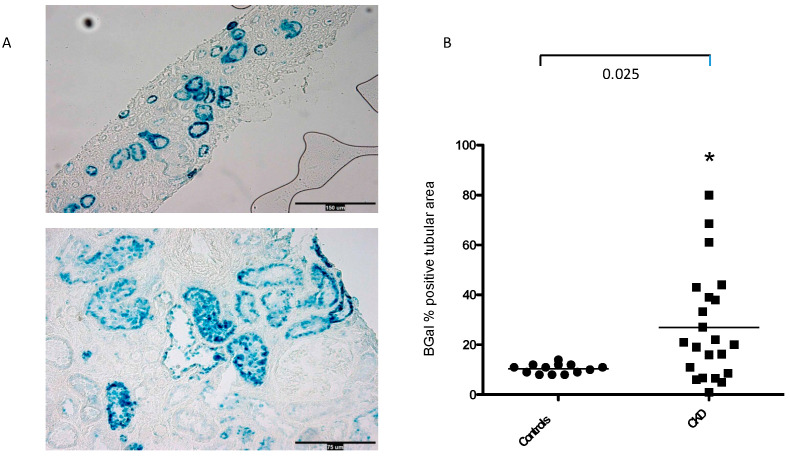
(**A**,**B**) Representative images of SA-β-Gal tubular expression in CKD patients. SA-β-Gal assay was ~2.6-fold higher [*p* < 0.025] in the tubular compartment of biopsies of CKD patients than of controls. Magnification × 200. SA-β-Gal, senescence-associated β-galactosidase; scale bar = 150–75 µm. * = *p* < 0.025.

**Figure 3 jcm-13-00322-f003:**
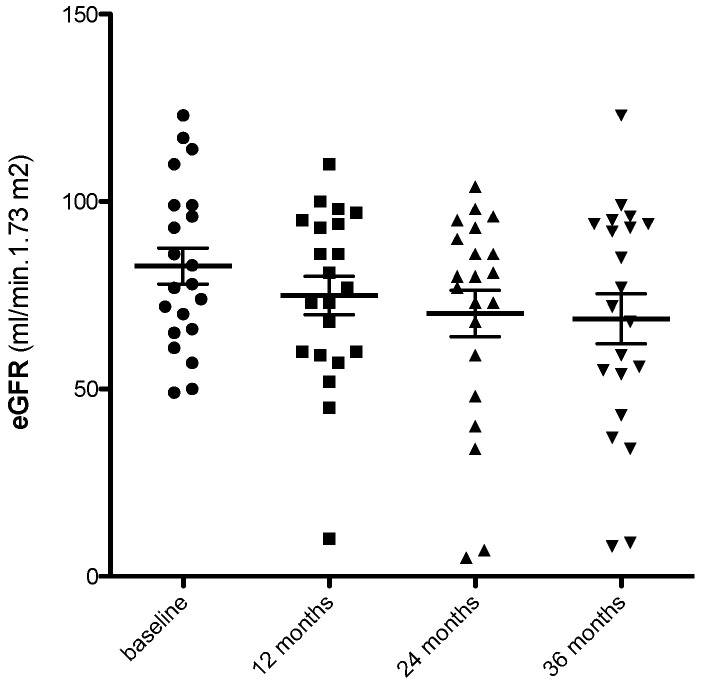
Estimated GFR (eGFR) during the study (means, 5–95th percentiles). Mean eGFR was 80 ± 5.7 mL/min/1.73 m^2^ at baseline, 75 ± 5 mL/min/1.73 m^2^ at year 1, 70 ± 6.1 mL/min/1.73 m^2^ at year 2, and 69 ± 6.7 mL/min/1.73 m^2^ at year 3. Mean change in eGFR from baseline to year 3 was −3.54 [95% CI, −0.598 to −6.48] mL/min/1.73 m^2^/year.

**Figure 4 jcm-13-00322-f004:**
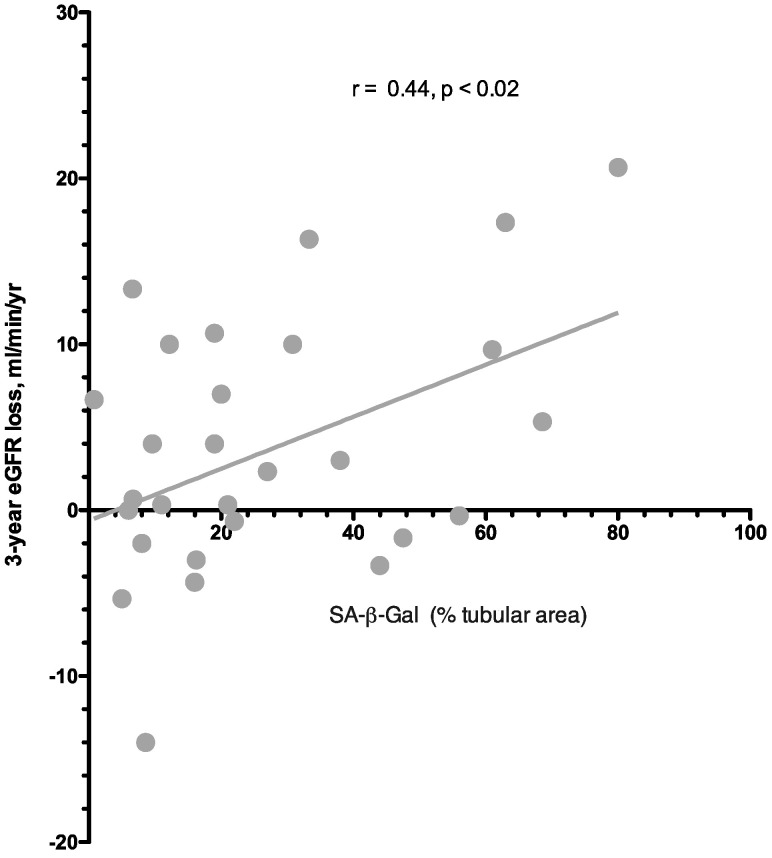
Relationship between SA-β-Gal expression in kidney tubules and eGFR loss at follow up (3 years).

**Figure 5 jcm-13-00322-f005:**
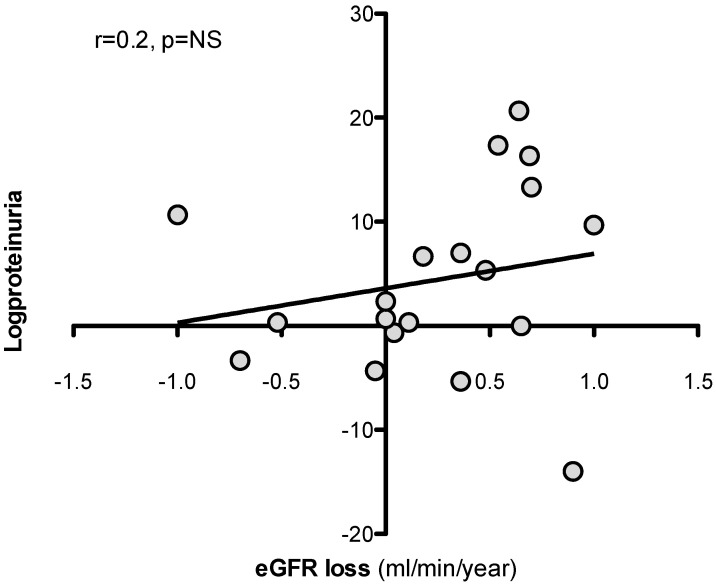
Relationship between logproteinuria and eGFR loss at follow up (3 years).

**Table 1 jcm-13-00322-t001:** Clinical characteristics of CKD patients.

Number of Subjects	22
Age (years)	50 ± 3
Gender (M/F)	9/13
BMI (Kg/m^2^)	28 ± 1.7
Proteinuria (g/day)	1.9 (0.75, 4.42)
Estimated GFR (mL/min 1.73 m^2^)	80 ± 5.7
eGFR loss (mL/min/year)	4.0 ± 1.93
Follow up (years)	3
SBP (mHg)	138 ± 7
DBP (mHg)	84 ± 4
Total Cholesterol (mg/dL)	222 (200, 259)
Triglycerides (mg/dL)	200 (100, 254)
HDL-C (mg/dL)	61 ± 5
LDL (mg/dL)	139 (84, 345)
IgA nephropathy (n)	5
Focal segmental glomerulosclerosis	5
membranous glomerulonephritis	5
diabetic kidney disease	2
hypertensive nephroangiosclerosis	2
minimal change disease	1
stage II lupus nephritis	1
chronic glomerulonephritis	1
RAAS blockade (n)	22
Statins	16
Steroids/immunosuppressive treatment	13

Data are mean ± SEM or median (IQR). Abbreviations: BMI = Body Mass Index, GFR = Glomerular Filtration Rate, SBP = Systolic Blood Pressure, DBP = Diastolic Blood Pressure, HDL-C = High-density Lipoprotein, LDL = Low-density Lipoprotein, RAAS = Renin Angiotensin Aldosterone System.

**Table 2 jcm-13-00322-t002:** Univariate analysis of the correlation between baseline clinical parameters and senescence-associated β-galactosidase (SA-β-Gal) expression in kidney tubules (*n* = 22).

Clinical Characteristics	SA-β-Gal (% Tubular Area)	
	R	*P*
Age (years)	0.08	NS
Serum creatinine (mg/dL)	0.05	NS
eGFR (mL/min.1.73 m^2^)	−0.28	NS
BMI (kg/m^2^)	0.10	NS
logProteinuria (g/day)	0.27	NS
logLDL-Cholesterol (mg/dL)	0.06	NS
logTriglycerides (mg/dL)	0.11	NS
logTotal Cholesterol	0.19	NS
SBP (mmHg)	0.15	NS
DBP (mmHg)	−0.09	NS

Abbreviations: eGFR = estimated glomerular filtration rate, BMI = body mass index, SBP = systolic blood pressure, and DBP = diastolic blood pressure. NS = Not Significant

**Table 3 jcm-13-00322-t003:** Univariate analysis of the correlation between senescence-associated β-galactosidase (SA-β-Gal), clinical parameters, and loss of eGFR at follow up (3 years) in patients with CKD (*n* = 22).

Parameter	eGFR Loss at 36 Months (mL/min.1.73 m^2^)	
	r	*p*
Age (years)	0.19	NS
logproteinuria (g/day)	0.20	NS
BMI (kg/m^2^)	0.09	NS
logLDL-Cholesterol (mg/dL)	0.01	NS
logTriglycerides (mg/dL)	0.003	NS
logTotal Cholesterol	0.06	NS
Immunosuppressive treatment	0.10	NS
SBP/DBP (mmHg)	0.1	NS

Abbreviations: eGFR = estimated glomerular filtration rate, BMI = body mass index, SBP = systolic blood pressure, and DBP = diastolic blood pressure. NS = Not Significant

## Data Availability

The authors are willing to make available individual data on request.

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
