# Peer review of "SA-β-Gal in Kidney Tubules as a Predictor of Renal Outcome in Patients with Chronic Kidney Disease"

_jcm, 2024, doi:10.3390/jcm13020322_

Round 1

Reviewer 1 Report

Comments and Suggestions for Authors

Review of the manuscript jcm-2766009

SA-β-Gal in kidney tubules as a predictor of renal outcome in patients with chronic kidney disease

by Pasquale Esposito et al.

The evaluated manuscript is an original paper. In the article, Authors tested the hypothesis that elevated lysosomal senescence-associated beta-galactosidase (SA-β-Gal) activity in kidney biopsies may be a predictor of subsequent eGFR loss in patients with chronic kidney disease. To answer the question, the Authors retrospectively studied the expression of SA-β-Gal in kidney biopsies obtained in a cohort of incident kidney patients who were followed up for 3 years as for the standard of care. The study revealed overexpression of SA-β-Gal in the tubular compartment of a large percentage of kidney biopsies at the time of diagnosis (63% of biopsies had >10% of tubuli expressing SA-β-Gal), a finding that suggests that an increase in SA--Gal activity in the tubular compartment is an early event in different kidney diseases. Moreover, the association between the overexpression of SA-β-Gal in the tubular compartment and eGFR loss at the follow-up was demonstrated, and the finding gave strength to the hypothesis that SA-β-Gal activity in kidney biopsies at the time of diagnosis is predictive of subsequent loss of function.

The concept of the study itself is very interesting, but I think that the authors should consider improving their manuscript. Some of the results have been presented unclearly.

1. Figure 1 – the number of excluded biopsy cases is 73. In the text below, the is an information (…) Sixty-six cases with eGFR<25 ml/min or insufficient tissue availability were excluded from further analysis (…) – it is unclear

2. Line 108 – 109 – Baseline clinical and demographic characteristics of subjects are shown in Table 1 – I cannot find the Table 1 in the text of the manuscript

3. My attention is drawn to the very high heterogeneity of the population of patients from whom kidney biopsies were taken (finally 22 biopsies were available, including IgA nephropathy [n=5], 7 FSGS [n=5], membranous glomerulonephritis [n=5], minimal change disease [n=1], chronic glomerulonephritis [n=1], hypertensive nephroangiosclerosis [n=2], stage II lupus nephritis [n=1], diabetic kidney disease [n=2].

Due to the various possible pathophysiological scenarios in these diseases, it is necessary to better justify why the authors believe that these patients can be treated as one common group - the results presented collectively do not assume differentiation into individual subpopulations (probably due to the too small number of individual subpopulations). In any case, it is a big methodological problem in my opinion.

4. Is it possible that as a result of thawing kidney tissue samples, physicochemical or biochemical disorders may occur, resulting in a change in the expression of the tested compound?

5. Patients from whom kidney samples were collected were treated with various drugs (single or double RAS blockade, statins, immunosuppressants). What specific medications were administered? Moreover, how can it be ruled out that the use of drugs does not affect the expression of the measured parameter in the kidneys?

It would be of special importance for angiotensin-converting enzyme inhibitors, due to their broad, pleiotropic and anti-inflammatory action profile. Thus, are there any reports that these drugs may affect the level of SA-β-Gal?

6. Figure 2 - control - the results of 9 people are marked - while the text of the manuscript states that the control group consisted of 13 people? (lines 106-107)

7. Table 2 – Correlation of SA-β-Gal expression and selected clinical parameters – one of the assessed parameters is eGFR. What eGFR is mentioned here, assessed at baseline or after 12, 24, 36 months of follow-up?

8. Table 3 – in the PDF version of the manuscript received for review the table is only partially legible? It is moved to the right side of the page - is it just two columns or more?

9. Furthermore, why are only proteinuria values reported in logarithm form in Tables 2 and 3?

Author Response

Reviewer 1

The evaluated manuscript is an original paper. In the article, Authors tested the hypothesis that elevated lysosomal senescence-associated beta-galactosidase (SA-β-Gal) activity in kidney biopsies may be a predictor of subsequent eGFR loss in patients with chronic kidney disease. To answer the question, the Authors retrospectively studied the expression of SA-β-Gal in kidney biopsies obtained in a cohort of incident kidney patients who were followed up for 3 years as for the standard of care. The study revealed overexpression of SA-β-Gal in the tubular compartment of a large percentage of kidney biopsies at the time of diagnosis (63% of biopsies had >10% of tubuli expressing SA-β-Gal), a finding that suggests that an increase in SA--Gal activity in the tubular compartment is an early event in different kidney diseases. Moreover, the association between the overexpression of SA-β-Gal in the tubular compartment and eGFR loss at the follow-up was demonstrated, and the finding gave strength to the hypothesis that SA-β-Gal activity in kidney biopsies at the time of diagnosis is predictive of subsequent loss of function.

The concept of the study itself is very interesting, but I think that the authors should consider improving their manuscript. Some of the results have been presented unclearly.

Question 1. Figure 1 – the number of excluded biopsy cases is 73. In the text below, the is an information (…) Sixty-six cases with eGFR<25 ml/min or insufficient tissue availability were excluded from further analysis (…) – it is unclear

Answer: We agree with the Reviewer. Thanks for checking this. In the revised version, the sentence has been corrected as follows:..” Seventy-three cases with eGFR<25 ml/min or insufficient tissue availability were excluded from further analysis”.

Question 2. Line 108 – 109 – Baseline clinical and demographic characteristics of subjects are shown in Table 1 – I cannot find the Table 1 in the text of the manuscript

 Answer: Thank you again. Table 1 was missing. It has now been added to the MS.

Question 3. My attention is drawn to the very high heterogeneity of the population of patients from whom kidney biopsies were taken (finally 22 biopsies were available, including IgA nephropathy [n=5], 7 FSGS [n=5], membranous glomerulonephritis [n=5], minimal change disease [n=1], chronic glomerulonephritis [n=1], hypertensive nephroangiosclerosis [n=2], stage II lupus nephritis [n=1], diabetic kidney disease [n=2].

Due to the various possible pathophysiological scenarios in these diseases, it is necessary to better justify why the authors believe that these patients can be treated as one common group - the results presented collectively do not assume differentiation into individual subpopulations (probably due to the too small number of individual subpopulations). In any case, it is a big methodological problem in my opinion.

 Answer: We agree with the reviewer in that the cohort appears very heterogeneous. However there is evidence that cell senescence is accelerated in different kidney diseases, including hypertensive disease (Hypertension. 52(1):123-9, 2008), AKI  (Am J Physiol Renal Physiol. 2010 298(5):F1078-94), IgA nephropathy (Transl Res. 2012;159(6):454–63), focal segmental glomerulosclerosis (Am J Nephrol. 51(12):950-958), diabetic nephropathy (Am J Physiol Renal Physiol. 295(5):F1563-73, 2008) chronic glomerulonephritis (Kidney Int 2007;71(3):218–26) etc. All these different disease have in common the upregulation of cell senescence markers in tubules, which correlates with IFTA, suggesting that kidney tubule senescence has common mechanisms underlying. Actually, a major issue is how to diagnose cell senescence in kidney biopsies. It has been proposed that SABgal can be used for screening, since this enzyme activity is upregulated by senescent in several different, other than kidney, tissues. However, scarce information is available on SABgal in human kidney biopsies.

According to the reviewer’s comments, the following sentence has included (Discussion):

Cell senescence is accelerated in several apparently different kidney diseases, including hypertensive disease, acute kidney injury (AKI), IgA nephropathy, focal segmental glomerulosclerosis, diabetic nephropathy and chronic glomerulonephritis [2,3], which suggests a common mechanism underlying.  SA-β-Gal staining has been proposed for initial cell senescence screening, since this enzyme activity is upregulated by senescence in several tissues. However, scarce information is available on SA-β-Gal in human kidney biopsies.

Question 4. Is it possible that as a result of thawing kidney tissue samples, physicochemical or biochemical disorders may occur, resulting in a change in the expression of the tested compound?

 Answer: We agree with the reviewer that  the enzymatic activity could be affected by fixative, cryopreservation protocol and storage time.

To perform the histochemical reaction at optimal assay conditions, we used a well-defined protocol. Fresh tissues were frozen  in optimal cutting temperature compound (OCT) using  cold isopentane and stored at -80°C.As described for many cutting-edge spatial analysis methods, fast controlled freezing preserves biological molecules, mimeses ice damage, and  embedding in OCT prepares tissues for future cryosectioning (Kenny Roberts, Liz Tuck 2019.Embedding and freezing fresh human tissue in OCT,  https://dx.doi.org/10.17504/protocols.io.66uhhew).  Moreover,  as previously reported β-gal activity staining decreases with cryopreservation time ( see a recent analysis, J Histochem Cytochem. 2020 Apr;68(4):269-278. doi: 10.1369/0022155420913534).  Therefore, for avoiding  an inaccurate evaluation, SA-b-Gal staining  was assessed 24 hours from the freezing of the sample.

The following sentence has been added to the methods, for better clarification:

SA-b-Gal staining was assessed 24 hours from the freezing of the sample.

Question 5. Patients from whom kidney samples were collected were treated with various drugs (single or double RAS blockade, statins, immunosuppressants). What specific medications were administered? Moreover, how can it be ruled out that the use of drugs does not affect the expression of the measured parameter in the kidneys? It would be of special importance for angiotensin-converting enzyme inhibitors, due to their broad, pleiotropic and anti-inflammatory action profile. Thus, are there any reports that these drugs may affect the level of SA-β-Gal?

Answer: Thank you for your question. Patients were treated according to clinical practice. Twenty-two subjects received single or double RAS blockade, and 16 subjects received statins (atorvastatin). Thirteen subjects received steroids and/or immunosuppressive therapy.  The stimulation of angiotensin type 1 (AT1) receptors, as well as of mineralocorticoid receptors, causes oxidative stress and vascular inflammation, which may lead to arterial stiffness and  vascular aging.Drugs of common practice in  nephrology, such as RAAS and mineralocorticoid blockade, atorvastatin, rosiglitazone and Omega-3 Fatty Acids are being re-examined for this purpose.

There are in vitro studies on senescence activated by RAS (Exp Mol Pathol. 2020 Dec;117:104551.

J Cell Physiol. 2021 Feb;236(2):1332-1344. etc) which was blunted, including SA-β-Gal staining by RAS blockade. We are not aware of in vivo human study on this.

Clearly, the issue you addressed is important and is a study limitation; this concept has been expressed in the Discussion:

…In our study we were not able to observe an association was between tubular SA-β-Gal expression and blood pressure levels; however almost all subjects were on treatment with RAAS inhibitors, which might have modified the expression or activity of SA-β-Gal. …

.. The cohort studied is small, and possibly statistically underpowered to detect all the existing clinical associations of tubular cell senescence.  In addition, virtually all patients received RAAS blockers and many were also on treatments with statins. Both drugs are known to posses a “senostatic” action, i.e. they can “downregulate” the senescent phenotype without removing senescent cells [30-32].

Question 6. Figure 2 - control - the results of 9 people are marked - while the text of the manuscript states that the control group consisted of 13 people? (lines 106-107).

 Answer: Thank you for checking this.There was partial overlapping of some points. Thirteeen cases actually can be individuated in Figure 2.

Question 7. Table 2 – Correlation of SA-β-Gal expression and selected clinical parameters – one of the assessed parameters is eGFR. What eGFR is mentioned here, assessed at baseline or after 12, 24, 36 months of follow-up?

Answer: Thank you for your question. Table 2 shows univariate analysis of the correlation between baseline eGFR and clinical parameters and SA-β-Gal expression in kidney tubules.

The term: ”baseline” has now been added to the caption.

Question 8. Table 3 – in the PDF version of the manuscript received for review the table is only partially legible? It is moved to the right side of the page - is it just two columns or more?

 Answer: We are sorry for this. Table 3 is composed by 2 columns. We hope that in the revised version there are no formatting issues.

Question 9. Furthermore, why are only proteinuria values reported in logarithm form in Tables 2 and 3?

Answer: Thank you for your question. While proteinuria was not normally distributed, SA-β-Gal passed normality tests (both KS normality tets and D’Agostino and Pearson). Accordingly, further analysis was made using logproteinuria. Accordingly, the following sentence has been added to the methods:

.. While proteinuria was not normally distributed SA beta Galactosidase passed normality tests. Accordingly, further analysis was made using logproteinuria.

In addition, in Table 2 and 3, we specified that cholesterol LDL cholesterol and triglycerides were log transformed before the analysis

Reviewer 2 Report

Comments and Suggestions for Authors

This is a retrospective analysis of SA beta Galactosidase staining in kidney tubules as a marker of senescence. The findings are interesting but retrospective nature of the study indicates that it could be hypothesis generating and not conclusive. This is further compromised by the 73 excluded and 20 lost to follow up. The varied etiologies in this small cohort could be confounders especially whether metabolic diseases and inflammatory components could promote this phenotype and whether amount of proteinuria and use of RAAS inhibitors may have an effect. Since CKD by its very nature obviously would have tubular senescence these markers would be overexpressed and the conclusion that they are a marker of progression may not be concluded based on this study design. Hence it appears that SA beta G is associated with tubular cell senescence and further conclusion that it accelerates progression of CKD may not be correct.

Author Response

Reviewer 2

Question. This is a retrospective analysis of SA beta Galactosidase staining in kidney tubules as a marker of senescence. The findings are interesting but retrospective nature of the study indicates that it could be hypothesis generating and not conclusive. This is further compromised by the 73 excluded and 20 lost to follow up.

Answer: We agree with the reviewer. In the study limitations the following sentence has been added.

.. However, our study has also several limitations. The study is retrospective and the cohort studied is small and possibly statistically underpowered to detect all the existing clinical associations of tubular cell senescence.  In addition, virtually all patients received RAAS and/or β-blockers and many were also on treatment with statins, drugs which are known to posses a “senostatic” action, i.e. they can “downregulate” the senescent phenotype [30-32]. For all these reasons findings obtained here need to be better considered as hypothesis-generating rather than conclusive, and need to be confirmed in multicentric, prospective, controlled studies.

Question. The varied etiologies in this small cohort could be confounders especially whether metabolic diseases and inflammatory components could promote this phenotype and whether amount of proteinuria and use of RAAS inhibitors may have an effect. Since CKD by its very nature obviously would have tubular senescence these markers would be overexpressed and the conclusion that they are a marker of progression may not be concluded based on this study design.

Answer: We agree with the reviewer on that the cohort is very heterogeneous. However there is evidence that cell senescence is accelerated in apparently different kidney diseases, including hypertensive disease (Hypertension.52(1):123-9, 2008), AKI  (Am J Physiol Renal Physiol. 2010 298(5):F1078-94), IgA nephropathy (Transl Res. 2012;159(6):454–63), focal segmental glomerulosclerosis (Am J Nephrol. 51(12):950-958), diabetic nephropathy (Am J Physiol Renal Physiol. 295(5):F1563-73, 2008) chronic glomerulonephritis (Kidney Int 2007;71(3):218–26) etc. All these different disease have in common the upregulation of cell senescence markers (mainly p16INKA and p27) in tubules, which correlates with IFTA, suggesting that kidney tubule senescence has common mechanisms underlying.  Less data are available on SABgal. Actually, a major issue is how to diagnose cell senescence in kidney biopsies. Currently, it is difficult to define unique markers which permit both the identification and quantification of senescent cells. Available studies are based on the presence/absence of cell-cycle arrest markers; however this may not be sufficient to confirm senescence, as the increase in nuclei positively immunostaining for p16INKA does not always coincide with the upregulation of p16INKA expression in vivo (Nat Med. 21(12):1424-35, 2015). It has been proposed that quantifying tissue senescence should comprise three approaches: staining of cell-cycle arrest markers, quantifying the effects of senescence- associated protein release, and excluding coexisting cellular proliferation (Nat Med. 21(12):1424-35, 2015). It has been proposed that SA-β-Gal  can be used for screening before applying the aforementioned approaches, since this enzyme activity is upregulated by senescent in several different, other than kidney, tissues. However, scarce information is available on SABgal in human kidney biopsies. Better than  focusing on cell senescence per se, our attention was SABgal. However we agree on the issues raised by the Reviewer.

According to the reviewer’s comments, the following sentence has included (Discussion):

Cell senescence is accelerated in several apparently different kidney diseases, including hypertensive disease, acute kidney injury (AKI), IgA nephropathy, focal segmental glomerulosclerosis, diabetic nephropathy and chronic glomerulonephritis [2,3], which suggests a common mechanism underlying. SA-β-Gal staining has been proposed for initial cell senescence screening, since this enzyme activity is upregulated by senescence in several tissues. However, scarce information is available on SA-β-Gal in human kidney biopsies.

In addition, in the limitations of the studied we added the following sentence:

..For all these reasons findings obtained here need to be better considered as hypothesis. generating than conclusive.

Question. Hence it appears that SA-β-Gal is associated with tubular cell senescence and further conclusion that it accelerates progression of CKD may not be correct.

Answer: We  agree with the Reviewer. However, we must say that the study was mainly focused on SA-β-Gal and not on senescence (as expressed by other markers). We agree on that  SA-β-Gal accelerates the progression of CKD cannot be ruled out by the study. Also in accordance with the Reviewer’s comments, the main conclusions have been modified as follows:

: .. SA-β-Gal is upregulated in the kidney tubular compartment of adult patients with CKD of different etiologies..

The sentence: “Therefore, our results reinforce the hypothesis that  SA-β-Gal overexpression contributes  to accelerated loss of eGFR in CKD..”  was changed into: …”Therefore, our results are in accordance with the hypothesis that  SA-β-Gal overexpression  is associated to accelerated loss of eGFR in CKD”.

Round 2

Reviewer 1 Report

Comments and Suggestions for Authors

Re-review of the manuscript jcm-2766009

SA-β-Gal in kidney tubules as a predictor of renal outcome in patients with chronic kidney disease

by Esposito P. et al.

Thank you for sending an updated version of the manuscript. The authors addressed  my comments and made some corrections to the current version of the manuscript.

However,  it is worth considering the following comments:

1. The .pdf file received for re-evaluation is again "poorly readable" in several places, e.g. page 7 - the Table "overlapping the text". The whole thing gives the impression of carelessness - it needs to be refined before eventual publication.

2. Figure 1 - it is better to put the number of rejected samples in the "horizontal" fields and the number of samples considered for analysis in the "vertical" fields. Hence, consistently, it would be better to enter the value 42 (115-73) in the middle horizontal field, and next to it introduce another horizontal field with the entry "lost to follow up n=20"

3. Table 1 – Clinical characteristics of patients – proteinuria, total cholesterol, triglycerides, LDL – results are given as median and values of the 3rd and 1st quartiles; the other parameters are expressed as mean ± SEM. Why are the values of these different parameters presented differently? This needs to be clarified. Why SEM and not SD?

4. Table 1 - why is there a "-" before the value in the DBD value?

5. Table 1 - it should also include information on how many patients had a given kidney disease and information on the medications used

6. "Study limitations" in the Conclusions should be more emphasized; the caveats we discussed in the first review should also be further mentioned there, especially in the context of the significant heterogeneity of samples from patients with many different nephrological conditions. It is also necessary to mention more about the potential impact of the drugs used, using the references indicated in the response to the first review in both cases. Consequently, the introduction of these new references will result in the need to re-numbering the items previously included in the manuscript.

Author Response

Re-review of the manuscript jcm-2766009

SA-β-Gal in kidney tubules as a predictor of renal outcome in patients with chronic kidney disease

by Esposito P. et al.

Thank you for sending an updated version of the manuscript. The authors addressed  my comments and made some corrections to the current version of the manuscript.

However,  it is worth considering the following comments:

Question 1. The .pdf file received for re-evaluation is again "poorly readable" in several places, e.g. page 7 - the Table "overlapping the text". The whole thing gives the impression of carelessness - it needs to be refined before eventual publication.

Answer:Thank you. We did our best to re.-format the revised version in the JCM format. However, we understand that it is not perfect. There are probably interactions with the JCM format and .pdf that we could not understand. We leave to the JCM protos the liberty to readjust the format in the final version, if any.

Question 2. Figure 1 - it is better to put the number of rejected samples in the "horizontal" fields and the number of samples considered for analysis in the "vertical" fields. Hence, consistently, it would be better to enter the value 42 (115-73) in the middle horizontal field, and next to it introduce another horizontal field with the entry "lost to follow up n=20"

Answer: Thank you. In the revised version we changed the Figure according to the Reviewer’s suggestion.

Question 3a. Table 1 – Clinical characteristics of patients – proteinuria, total cholesterol, triglycerides, LDL – results are given as median and values of the 3rd and 1st quartiles; the other parameters are expressed as mean ± SEM. Why are the values of these different parameters presented differently? This needs to be clarified. 

Answer: Thank you for your question. Cholesterol, triglycerides, and LDL results were skewed.  

In the revised version we have specified this (methods):

..Parameters not normally distributed were logarithmically transformed for statistical analysis. While proteinuria,cholesterol, triglycerides and LDL were not normally distributed, SA-β-Gal passed normality tests…

Question 3b Why SEM and not SD?

Answer: As known, both the SD and the  SEM are used to present the characteristics of sample data and explain statistical analysis results. In our previous studies on cell loss in the kidney (Refs 13,16,17) SEM has always been used, as suggested by a previous editorial comment, as SEM measures how far the sample mean of the data is likely to be from the true population mean. For this reason, in the present study we kept the use of SEM. However, this simply represents homogeinity with previous studies. We can move to SD if it is necessary.

Question 4. Table 1 - why is there a "-" before the value in the DBD value?

Answer:Thank you for checking this. The sign "-"  has been removed.

Question 5. Table 1 - it should also include information on how many patients had a given kidney disease and information on the medications used

Answer: In the revised Table 1 we added the information requested.

Question 6. "Study limitations" in the Conclusions should be more emphasized; the caveats we discussed in the first review should also be further mentioned there, especially in the context of the significant heterogeneity of samples from patients with many different nephrological conditions. It is also necessary to mention more about the potential impact of the drugs used, using the references indicated in the response to the first review in both cases. Consequently, the introduction of these new references will result in the need to re-numbering the items previously included in the manuscript.

Answer: Thank you for your suggestions. Five new references have been added. In addition the conclusions have been changed as follows:

…In summary, our study shows that SA-β-Gal is upregulated in the kidney tubular compartment of adult patients with CKD of different etiologies, and that SA-β-Gal expression >30% in kidney tubuli is associated with accelerated loss of renal function. Therefore, our results are in accordance with the hypothesis that SA-β-Gal overexpression is associated to accelerated loss of eGFR in CKD. This study provides a clinical basis for the use of SA-β-Gal as a marker of kidney senescence in CKD, a common clinical condition that is still associated with an unfavorable prognosis. However our results cast some doubt on the use of SA-β-Gal as a unique screening marker of cell senescence in the human kidney. Our study has several strenghts, since it includes a real-life renal biopsy sample cohort from incident patients with quite long follow-up. Even if the cohort studied may appear to be heterogeneous, it is important underlining that cell senescence is accelerated in apparently different kidney diseases, including hypertensive disease [24], AKI [30], IgA nephropathy {31], focal segmental glomerulosclerosis [16], diabetic nephropathy [13] and chronic glomerulonephritis [32]. All these different diseases have in common the upregulation of cell senescence markers (mainly p16INKA and p27) in tubules, which correlates with IFTA, suggesting that kidney tubule senescence has common mechanisms underlying. However, our study has several limitations. The study is retrospective and the cohort studied is small and possibly statistically underpowered to detect all the existing clinical associations of tubular cell senescence. In addition, all patients received RAAS and/or β-blockers and many were also on treatment with statins, drugs which are known to posses a “senostatic” action, i.e. they can “downregulate” the senescent phenotype [33-37]. The stimulation of angiotensin type 1 (AT1) receptors, as well as of mineralocorticoid receptors, causes oxidative stress and vascular inflammation, which may lead to arterial stiffness and vascular aging [36,37]. Drugs of common practice in nephrology, such as RAAS and mineralocorticoid blockade, atorvastatin, rosiglitazone and Omega-3 Fatty Acids are being re-evaluated for their anti-senescence action. For all these reasons findings obtained here need to be better considered as hypothesis-generating rather than conclusive, and need to be confirmed in multicentric, prospective, controlled studies.

Thank you so much for constructive criticism. giacomo garibotto